# A field test of the dilution effect hypothesis in four avian multi-host pathogens

**Martina Ferraguti**[1¤a], **Josué Martínez-de la Puente**[1,2¤b], **Miguel Ángel Jiménez–Clavero**[2,3], **Francisco Llorente**[3], **David Roiz**[1¤c], **Santiago Ruiz**[2,4], **Ramón Soriguer**[2,5], **Jordi Figuerola**[1,2]*

**1** Department of Wetland Ecology, Doñana Biological Station (EBD–CSIC), Seville, Spain, **2** CIBER of Epidemiology and Public Health (CIBERESP), Madrid, Spain, **3** Centro de Investigación en Sanidad Animal, Instituto Nacional de Investigación y Tecnología Agraria y Alimentaria (INIA–CISA), Valdeolmos, Madrid, Spain, **4** Diputación de Huelva, Área de Medio Ambiente, Servicio de Control de Mosquitos, Huelva, Spain, **5** Department of Ethology & Biodiversity Conservation, Doñana Biological Station (EBD–CSIC), Seville, Spain

¤a Current address: Department of Theoretical and Computational Ecology (TCE), Institute for Biodiversity and Ecosystem Dynamics (IBED), University of Amsterdam, Amsterdam, The Netherlands
¤b Current address: Department of Parasitology, University of Granada (UGR), Granada, Spain
¤c Current address: MIVEGEC, University Montpellier, IRD, CNRS, Montpellier, France
* jordi@ebd.csic.es

**Data Availability Statement:** All relevant data, including the complete dataset, are within the manuscript and its Supporting Information files.

## Abstract

The Dilution Effect Hypothesis (DEH) argues that greater biodiversity lowers the risk of disease and reduces the rates of pathogen transmission since more diverse communities harbour fewer competent hosts for any given pathogen, thereby reducing host exposure to the pathogen. DEH is expected to operate most intensely in vector-borne pathogens and when species-rich communities are not associated with increased host density. Overall, dilution will occur if greater species diversity leads to a lower contact rate between infected vectors and susceptible hosts, and between infected hosts and susceptible vectors. Field-based tests simultaneously analysing the prevalence of several multi-host pathogens in relation to host and vector diversity are required to validate DEH. We tested the relationship between the prevalence in house sparrows (*Passer domesticus*) of four vector-borne pathogens– three avian haemosporidians (including the avian malaria parasite *Plasmodium* and the malaria-like parasites *Haemoproteus* and *Leucocytozoon*) and West Nile virus (WNV)–and vertebrate diversity. Birds were sampled at 45 localities in SW Spain for which extensive data on vector (mosquitoes) and vertebrate communities exist. Vertebrate censuses were conducted to quantify avian and mammal density, species richness and evenness. Contrary to the predictions of DEH, WNV seroprevalence and haemosporidian prevalence were not negatively associated with either vertebrate species richness or evenness. Indeed, the opposite pattern was found, with positive relationships between avian species richness and WNV seroprevalence, and *Leucocytozoon* prevalence being detected. When vector (mosquito) richness and evenness were incorporated into the models, all the previous associations between WNV prevalence and the vertebrate community variables remained unchanged. No significant association was found for *Plasmodium* prevalence and vertebrate community variables in any of the models tested. Despite the studied system having several characteristics that should favour the dilution effect (i.e., vector-borne pathogens,

**Funding:** This study was funded by project P11-RNM-7038 from the Junta de Andalucía and project PGC2018-095704-B-100 from the Spanish Ministry of Science and Innovation and European (FEDER) funds to JF. MF is currently funded by the Marie Sklodowska-Curie Fellowship from the European Commission (grant number 844285, 'EpiEcoMod'). JMP was partially supported by a 2017 Leonardo Grant for Researchers and Cultural Creators, BBVA Foundation. The funders had no role in study design, data collection and analysis, decision to publish, or preparation of the manuscript.

**Competing interests:** The authors have declared that no competing interests exist.

an area where vector and host densities are unrelated, and where host richness is not associated with an increase in host density), none of the relationships between host species diversity and species richness, and pathogen prevalence supported DEH and, in fact, amplification was found for three of the four pathogens tested. Consequently, the range of pathogens and communities studied needs to be broadened if we are to understand the ecological factors that favour dilution and how often these conditions occur in nature.

## Author's summary

The Dilution Effect Hypothesis (DEH) postulates that biodiversity can reduce disease epidemics because more diverse communities harbour a lower fraction of competent hosts, which thus reduces pathogen prevalence. Here, we tested DEH by using field information from 45 populations in SW Spain on the prevalence of four vector-borne pathogens and considered both the potential role of the vertebrate community and mosquito vectors. We determined the prevalence of *Plasmodium*, *Haemoproteus*, *Leucocytozoon* and antibodies for the zoonotic West Nile virus in wild house sparrows. Contrary to the predictions of DEH, our results do not support the general protective ability of biodiversity to reduce the prevalence of these four pathogens.

## Introduction

The number of emerging infectious diseases affecting humans is currently increasing [1] and approximately 75% of such diseases are known to be of zoonotic origin [2]. Many are caused by vector-borne pathogens that potentially have detrimental effects on human populations and cause serious concerns for public health [3]. The magnitude of this problem became apparent when the reported number of vector-borne diseases in the period 2004–2016 in the United States doubled [4]. The Dilution Effect Hypothesis (DEH) argues that biodiversity is related to reduced pathogen prevalence because species-rich communities harbour a lower fraction of competent hosts (i.e., individuals in which the pathogen can multiply to sufficient levels to pass the infection onto a new susceptible individual), which thus reduces pathogen transmission success and, consequently, pathogen prevalence [5,6]. In the case of vector-borne pathogens, in more diverse communities a higher fraction of vector bites is expected to occur on non-competent hosts that 'dilute' the pathogens in the community, thereby reducing both pathogen prevalence in vectors and the number of susceptible hosts [7]. However, theoretical models suggest that both negative (dilution) and positive (amplification) relationships between pathogen prevalence and biodiversity occur [8,9] and, indeed, both phenomena have been observed in wild populations. For example, the dilution effect was reported by Swaddle et al. [10], who noted a lower incidence of West Nile virus (WNV) in humans in US counties with richer avian (i.e., the vertebrate reservoirs of the virus) communities. Conversely, an amplification effect was reported by Roiz et al. [11] in a study in SW Spain, where a higher prevalence of Usutu virus was found in areas with richer avian communities and, in particular, in areas with more passerine species. Given its implications for public health, the validity and generality of the relationship between biodiversity and pathogen prevalence suggested by DEH has been the focus of intense research efforts in recent years [8,12,13]. As support for DEH, negative relationships between host species richness and pathogen prevalence have been reported in several pathogens transmitted by ticks (e.g., Lyme disease [14,15]), mosquitoes (e.g., WNV [10]) and

rodents (e.g., hantavirus [16]). However, isolating the effects of host community composition or of the presence of a highly competent species for the pathogen in question is difficult [12,17]. In one example, Kilpatrick et al. [18] demonstrated that American robins (*Turdus migratorius*) were responsible for most WNV-infectious mosquitoes and as such acted as super-spreaders. Indeed, in most of these previous studies, pathogen prevalence was related to species richness (i.e., the number of different species in the area) rather than species diversity, which takes into account the relative proportion of the different species present in the area (e.g., Shannon, Simpson or other evenness indices). The relationship between species richness and/or diversity and pathogen prevalence may be due either to the presence of key species or to the identity of the species included in the community and its density, rather than to any intrinsic property of biodiversity [19]. As well, sample bias between locations can influence the estimation of richness, an issue that could be solved by controlling for differences in the number of individuals and the number of samples collected (i.e., using rarefaction approaches) [20]. Additionally, Johnson et al. [21] proposed that dilution should be more evident at local scales but weaker at larger scales, since biotic interactions occur locally while abiotic factors tend to dominate at larger scales (see also [22]). Fundamentally, DEH occurs in association with an increase in species diversity leading to a decrease in the relative density of susceptible host density, thereby reducing contact rates between pathogen vectors and susceptible hosts in the case of vector-borne pathogens. By contrast, amplification can occur when increased diversity leads to the opposite phenomenon [23]. In addition, various authors have also suggested that more diverse host communities may harbour a higher number of host individuals, which could help vectors proliferate and, eventually, could increase pathogen transmission, thereby neutralizing any potential dilution effect [12]. However, the density of vectors and their distributions are traditionally linked to landscape and climate [24], and the relevance of host abundance and distribution to the distribution and abundance of vectors remains poorly known [12]. Recently, Rohr et al. [25] evaluated the conditions that facilitate a negative relationship between biodiversity and pathogen transmission. These authors concluded that the dilution effect is more likely to occur in vector-borne pathogens and will be largely influenced by community assembly rules. In particular, dilution effects may be expected to occur more often when the community assembly is substitutive as opposed to additive. In the latter case, the increase in the number of species is associated with increases in host densities since the individuals of the new species are simply added to those of the species that are already present. When community assembly is substitutive, however, an increase in species richness does not translate into an increase in the number of hosts. Nevertheless, only a few studies have ever tested all these relationships under natural conditions and consequently empirical tests are still urgently required [12]. Unfortunately, no detailed information is available on the host competence of the avian species present in southern Spain. Consequently, it is not possible to analyse how community competence varies within the assemblage, although it is possible to analyse whether or not vector and vertebrate community richness and density fit better in additive or substitutive assemblage models.

Civitello et al. [26] undertook a meta-analysis that found support for DEH in a wide range of host-parasite systems including pathogens with different transmission pathways, lifecycles, and host ranges, in both parasites that affect humans and others that only infect wildlife. Nevertheless, this meta-analysis has been criticized because it did not consider the negative publication bias; many of the studies included in this review were performed under simplified laboratory conditions in which individuals of a competent and a non-competent species were combined in an artificial mesocosm, which raises doubts about the validity of their conclusions when applied to multi host-pathogen systems [27]. A similar meta-analysis based on field studies of public-health-relevant pathogens failed to find support for the dilution effect [28]. Given

such a variety of outcomes, a potentially reliable approach for testing DEH could come from studying pathogens circulating in a single ecosystem, which would assist in accurately separating effects derived from biodiversity from those originating due to differences in host and vector community composition [8]. Vector diversity has been traditionally ignored in most of the studies focusing on DEH, although such information is essential for validating this hypothesis given that pathogen incidence is largely determined by vector distribution [12,29]. Indeed, Roche et al. [29] elaborated a multi-species Susceptible-Infectious-Recovered transmission model and concluded that increased vertebrate host-species richness decreased WNV transmission, while vector species-richness increased pathogen transmission. This model was built on the assumptions that in both vertebrate and vector communities the most abundant host reservoirs and vectors had the highest susceptibility for pathogen transmission. In other words, species rich communities are created by adding individuals of non-susceptible hosts or vector species to species poor communities, thereby hindering the spread of transmission.

Here, we investigated the effects of both mosquito and vertebrate community characteristics on the transmission of four vector-borne avian pathogens in order to test the predictions of DEH in a natural system. We sampled wild house sparrows (*Passer domesticus*) as susceptible hosts for avian malaria parasites and related haemosporidians belonging the genera *Plasmodium*, *Haemoproteus* and *Leucocytozoon* [30], and the flavivirus WNV [31]. While mosquitoes transmit WNV and *Plasmodium* parasites, *Haemoproteus* (subgenera *Parahaemoproteus*) is transmitted by *Culicoides* biting midges and *Leucocytozoon* by black flies [32]. These four widely distributed pathogens infect wild birds [32,33] but only WNV is also able to induce disease in mammals. In fact, mammals are dead-end hosts of WNV [33] as they do not develop sufficient viremia levels when infected to transmit the virus to the mosquitoes that feed on them. A basic premise of DEH is that the host competence for each pathogen varies between species, which is the case of the four pathogens studied here. For instance, compared to *Haemoproteus* and *Leucocytozoon* parasites, *Plasmodium* spp. are considered generalist parasites infecting birds of different taxa (i.e., orders) [34], although differences may occur between parasite lineages [35] and owing to environmental conditions [36]. In addition, although WNV is a generalist pathogen with a complex eco-epidemiology that is known to replicate in more than 300 species of birds (https://www.cdc.gov/westnile/dead-birds/index.html), there is important interspecific variation in host competence due to differences in the magnitude and duration of the peak of the viremia [37].

Thus, the aims of our study were to analyse the prevalence of these four avian pathogens in 45 sparrow populations and determine whether or not the DEH best explains the observed infection patterns. Specifically, we would expect a lower prevalence of the four studied pathogens as vertebrate species richness or diversity increased. We used different metrics of species richness and diversity in vector and host communities to understand their potential effect on the occurrence of DEH. In particular, in the case of species richness, we estimated the rarefaction curve (for the sake of clarity, hereafter referred to as 'richness') [20,38] to take into account the differences between localities in the number of samples taken. Additional models were also fitted using the raw number of different species registered at each locality. As a diversity measure, to estimate the evenness we used Shannon's equitability index [39], while for avian hosts we also estimated avian phylogenetic diversity [40,41]. By analysing different metrics of species diversity and richness, we ensured that the results reported in this study are robust to variable selection and do not depend on which parameters are selected to estimate richness and diversity. Here, we report the results of the analyses using the number of species projected from a rarefaction curve to estimate richness, and evenness as a measure of diversity. The results from models performed with the raw number of different species in each locality

and the avian phylogenetic diversity did not differ quantitatively (see Supporting Information) and are only commented on in the main text when qualitative differences were found.

Finally, we tested the main relationships between different biodiversity components of hosts and vectors that have been proposed as favouring or limiting the dilution effect in the wild. First, we determined whether or not the assembly of vectors and hosts followed an additive community model. To do so, we tested for a positive relationship between vector species richness and vector density, and for a positive relationship between avian species richness and avian diversity, as expected under an additive model of community assembly. This additive model was assumed for vector and host communities in the model created by Roche et al. [29] that supports the dilution effect; however, it was considered by Rohr et al. [25] to be a mechanism that could limit the dilution effect in hosts (see also [21]). Secondly, we tested the idea that denser host communities will support denser vector communities–thereby reducing the potential for disease dilution [12]–by testing the relationship between avian and mammal (host) densities and vector density, a relationship that is often taken for granted but for which to date there is little empirical evidence.

## Results

### Pathogen prevalence and host and vector community association

Data on infection by *Plasmodium*, *Haemoproteus* and *Leucocytozoon* parasites was taken from 2,588 house sparrows (range: 10–105 individuals per locality); West Nile virus seroprevalence was recorded for 2,544 of these sparrows (range: 10–102).

The prevalence of *Plasmodium*, *Haemoproteus* and *Leucocytozoon* and the seroprevalence of WNV were 29.6% (95% C.I.: 27.8–31.3), 14.1% (95% C.I.: 12.8–15.5), 28.6% (95% C.I.: 26.9–30.4) and 0.7% (95% C.I.: 0.4–1.1), respectively. The results from the models including only vertebrate-related variables are summarized in Table 1. Infection by WNV and *Leucocytozoon*

**Table 1. Results of the GLMMs testing the relationships between the prevalence of avian malaria *Plasmodium*, the related *Haemoproteus* and *Leucocytozoon* parasites (N = 2,588), and the seroprevalence of WNV (N = 2,544), and individual characteristics of house sparrows (age, sex, and month of capture), avian and mammal species density, richness (estimated from a rarefaction curve) and diversity (calculated as evenness index).** Significant relationships ($p \leq 0.05$) are highlighted in bold; conditional and marginal (in brackets) $R^2$ variance are shown.

| Independent variable | *Plasmodium* Estimate (±S.E.) | χ2 | d.f. | p | *Haemoproteus* Estimate (±S.E.) | χ2 | d.f. | p | *Leucocytozoon* Estimate (±S.E.) | χ2 | d.f. | p | West Nile virus Estimate (±S.E.) | χ2 | d.f. | p |
|---|---|---|---|---|---|---|---|---|---|---|---|---|---|---|---|---|
| Intercept | 0.56 (0.99) | 0.32 | 1 | 0.57 | 0.01 (1.44) | 0.00 | 1 | 0.99 | -1.42 (1.15) | 1.53 | 1 | 0.22 | 0.99 (4.16) | 0.05 | 1 | 0.81 |
| Month | -0.13 (0.06) | 4.35 | 1 | **0.04** | -0.19 (0.09) | 3.62 | 1 | 0.06 | -0.12 (0.07) | 2.53 | 1 | 0.11 | -0.82 (0.30) | 7.77 | 1 | **0.005** |
| Sex: male | 0.00[a] | 0.19 | 1 | 0.66 | 0.00[a] | 5.17 | 1 | **0.02** | 0.00[a] | 1.31 | 1 | 0.25 | 0.00[a] | 0.34 | 1 | 0.56 |
| Sex: female | 0.04 (0.09) | | | | -0.32 (0.14) | | | | -0.12 (0.10) | | | | -0.24 (0.41) | | | |
| Age: unknown | 0.00[a] | 5.87 | 2 | **0.05** | 0.00[a] | 20.92 | 2 | **<0.001** | 0.00[a] | 37.31 | 2 | **<0.001** | 0.00[a] | 1.46 | 2 | 0.48 |
| Age: juvenile | -0.17 (0.15) | | | | -0.71 (0.29) | | | | 0.04 (0.18) | | | | -0.43 (1.08) | | | |
| Age: adult | -0.44 (0.19) | | | | -0.03 (0.33) | | | | 0.86 (0.22) | | | | 0.07 (1.13) | | | |
| Avian density | -0.01 (0.01) | 2.56 | 1 | 0.11 | -0.01 (0.01) | 0.98 | 1 | 0.32 | -0.01 (0.01) | 0.87 | 1 | 0.37 | 0.00 (0.01) | 0.01 | 1 | 0.94 |
| Avian richness | -0.02 (0.03) | 0.60 | 1 | 0.44 | 0.06 (0.04) | 2.15 | 1 | 0.14 | 0.08 (1.31) | 4.96 | 1 | **0.02** | 0.37 (0.11) | 12.18 | 1 | **<0.001** |
| Avian diversity | 0.95 (1.05) | 0.81 | 1 | 0.37 | -0.53 (1.56) | 0.12 | 1 | 0.73 | -0.09 (1.31) | 0.01 | 1 | 0.95 | -5.68 (4.14) | 1.88 | 1 | 0.17 |
| Mammal density | -0.01 (0.01) | 0.57 | 1 | 0.45 | -0.05 (0.02) | 6.97 | 1 | **0.01** | 0.06 (0.02) | 12.01 | 1 | **<0.001** | -0.05 (0.05) | 1.24 | 1 | 0.26 |
| Mammal richness | 0.06 (0.02) | 0.14 | 1 | 0.71 | -0.53 (0.25) | 4.64 | 1 | **0.03** | -0.25 (0.21) | 1.35 | 1 | 0.24 | -2.59 (0.61) | 17.85 | 1 | **<0.001** |
| Mammal diversity | -0.90 (0.50) | 3.24 | 1 | 0.07 | 1.22 (0.55) | 4.97 | 1 | **0.02** | 0.47 (0.52) | 0.79 | 1 | 0.37 | 7.83 (1 84) | 18.04 | 1 | **<0.001** |
| $R^2$ (%) | 3.72 (15.40) | | | | 9.74 (37.72) | | | | 7.39 (36.83) | | | | 49.34 (69.98) | | | |

a Reference category.

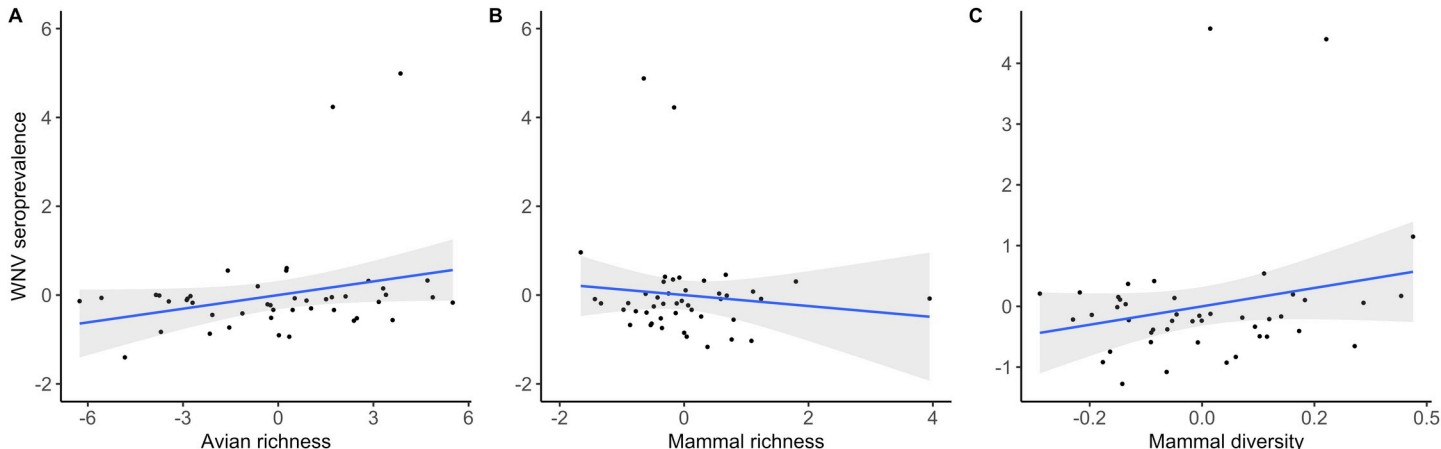

**Fig 1.** Leverage plots showing the relationships between WNV seroprevalence in house sparrows from the 45 localities included in this study and A) avian richness (estimated from a rarefaction curve), B) mammal richness (estimated from a rarefaction curve), and C) mammal diversity (measured as the evenness index). The prevalence of *Leucocytozoon* was calculated using the least squares means of a GLM controlling for birds' ages and locality. The 95% confidence level interval is shown in grey.

were positively associated with avian species richness (Figs 1A and 2A). In the case of variables reflecting the mammal communities, different significant associations were found. Infection by *Haemoproteus* was negatively associated with mammal density (Fig 3A), an association that was positive in the case of *Leucocytozoon* (Fig 2B). Both infection by *Haemoproteus* and WNV seroprevalence were negatively associated with mammal richness (Figs 1B and 3B) but positively related with mammal diversity (Figs 1C and 3C).

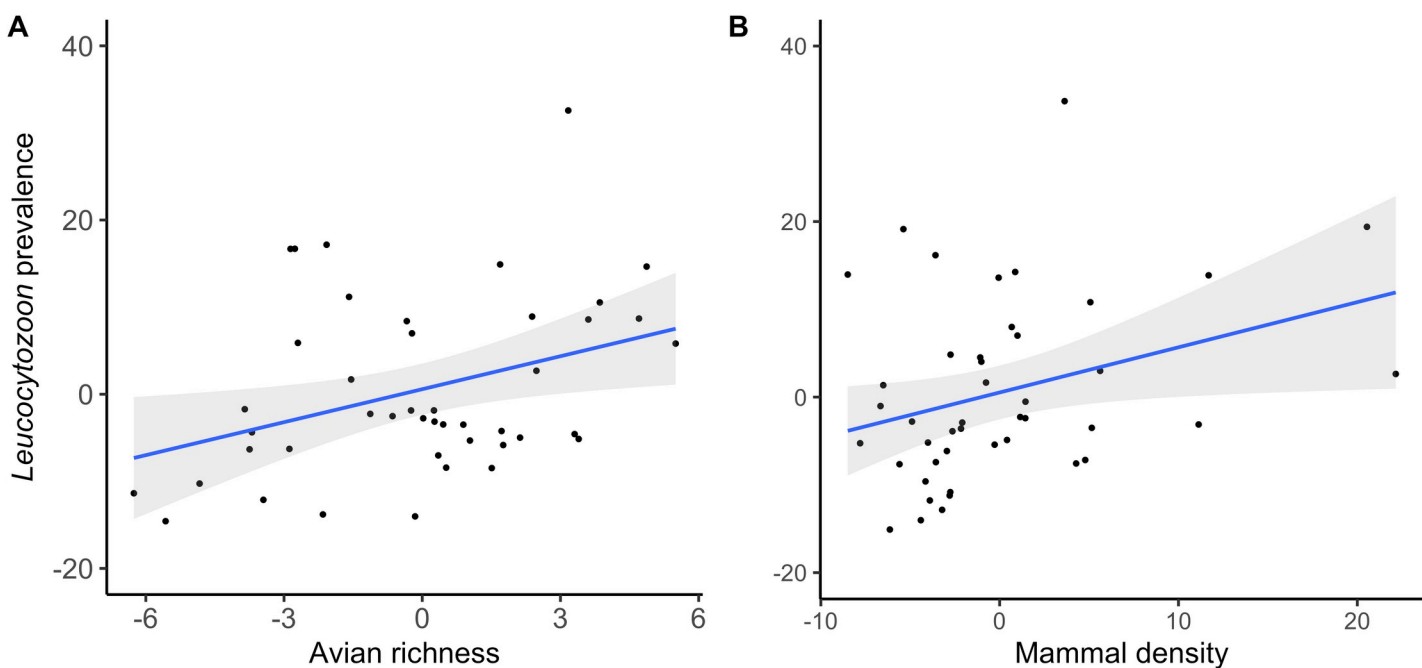

**Fig 2.** Leverage plots showing the relationships between *Leucocytozoon* prevalence in house sparrows from the 45 localities included in this study and A) avian richness (estimated from a rarefaction curve) and B) mammal density (sum of the densities of all mammals detected at each locality). The prevalence of *Leucocytozoon* was calculated using the least squares means of a GLM controlling for birds' ages and locality. The 95% confidence level interval is shown in grey.

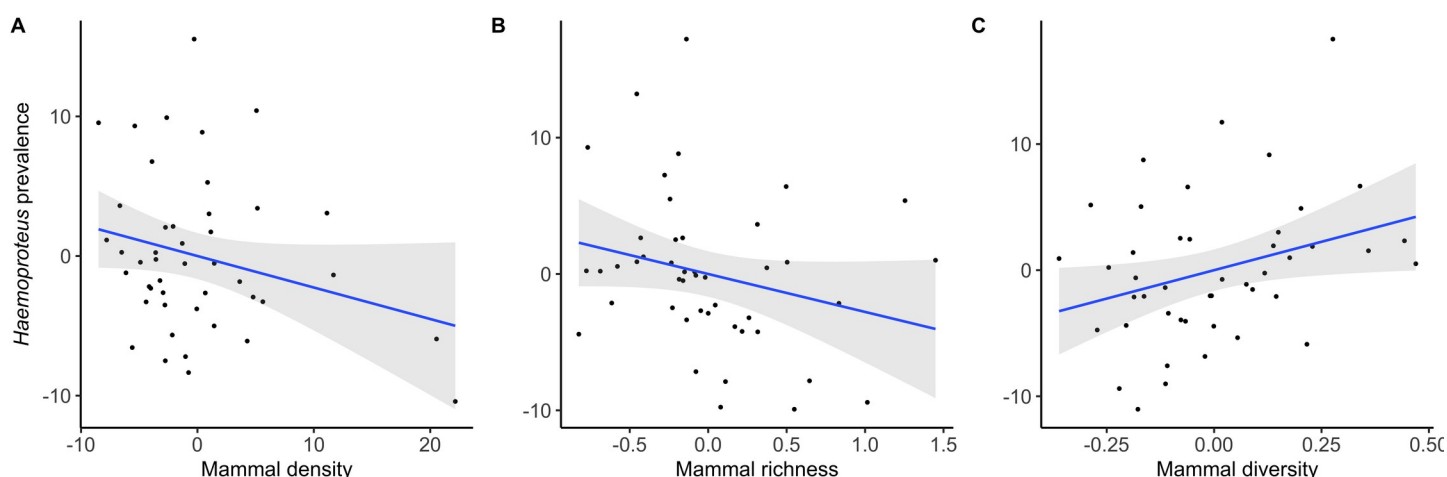

**Fig 3.** Leverage plots showing the relationships between *Haemoproteus* prevalence in house sparrows from the 45 localities included in this study and A) mammal density (sum of the density of all mammal species detected at each locality), B) mammal richness (estimated from a rarefaction curve) and C) mammal diversity (measured as the evenness index). The prevalence of *Haemoproteus* calculated using the least squares means of a GLM controlling for birds' sex and age, and locality. The 95% confidence level interval is shown in grey.

When mosquito-related variables were added to *Plasmodium* and WNV models (the two studied mosquito-borne pathogens), all the previous significant associations between pathogen infections and variables reflecting vertebrate communities remained the same (Table 2). No significant association was found between infection by *Plasmodium* or WNV and either mosquito species richness or diversity (Table 2).

**Table 2. Results of the GLMMs analysing the relationship between the prevalence of the two studied mosquito-borne pathogens: avian malaria *Plasmodium* (N = 2,588) and seroprevalence of WNV (N = 2,544) and the individual characteristics of the house sparrows (age, sex, and month of capture), avian and mammal species density, richness (measured from a rarefaction curve) and diversity (calculated as evenness index), vector richness (estimated from a rarefaction curve) and diversity (calculated as evenness index).** Significant relationships ($p \leq 0.05$) are highlighted in bold. Conditional and marginal (in brackets) $R^2$ variance are shown.

| Independent variable | *Plasmodium* Estimate (±S.E.) | χ2 | d.f. | *p* | West Nile virus Estimate (±S.E.) | χ2 | d.f. | *p* |
|---|---|---|---|---|---|---|---|---|
| Intercept | 1.35 (1.18) | 1.32 | 1 | 0.25 | 0.06 (4.63) | 0.00 | 1 | 0.99 |
| Month | -0.12 (0.03) | 3.70 | 1 | **0.05** | -0.84 (0.30) | 7.97 | 1 | **0.005** |
| Sex: male | 0.00[a] | 0.16 | 1 | 0.69 | 0.00[a] | 0.32 | 1 | 0.57 |
| Sex: female | 0.04 (0.09) | | | | -0.23 (0.41) | | | |
| Age: unknown | 0.00[a] | 5.44 | 2 | 0.06 | 0.00[a] | 1.63 | 2 | 0.44 |
| Age: juvenile | -0.16 (0.15) | | | | -0.41 (1.07) | | | |
| Age: adult | -0.43 (0.19) | | | | 0.14 (1.13) | | | |
| Avian density | -0.01 (0.01) | 1.17 | 1 | 0.28 | 0.01 (0.01) | 0.14 | 1 | 0.71 |
| Avian richness | 0.01 (0.04) | 0.01 | 1 | 0.94 | 0.37 (0.11) | 10.68 | 1 | **0.001** |
| Avian evenness | 0.68 (1.09) | 0.39 | 1 | 0.53 | -4.21 (4.69) | 0.80 | 1 | 0.37 |
| Mammal density | 0.01 (0.02) | 0.01 | 1 | 0.91 | -0.07 (0.05) | 1.68 | 1 | 0.19 |
| Mammal richness | -0.07 (0.19) | 0.12 | 1 | 0.72 | -2.17 (0.89) | 5.88 | 1 | **0.001** |
| Mammal diversity | -0.16 (0.65) | 0.07 | 1 | 0.79 | 7.01 (2.45) | 8.12 | 1 | **0.004** |
| Mosquito richness | -0.17 (0.17) | 1.01 | 1 | 0.31 | -0.46 (0.39) | 1.39 | 1 | 0.23 |
| Mosquito diversity | -1.05 (0.83) | 1.61 | 1 | 0.20 | 3.24 (3.51) | 0.85 | 1 | 0.35 |
| $R^2$ (%) | 3.86 (14.36) | | | | 44.88 (63.10) | | | |

a Reference category.

Additionally, relationships between vertebrate communities and pathogen infection were, with few exceptions, qualitatively the same in models that include the raw number of different avian species registered at each locality and the avian phylogenetic diversity (S1 and S3 Tables). These exceptions were a positive association between *Haemoproteus* prevalence and avian richness (S1 and S3 Tables), and a negative relationship between *Plasmodium* prevalence and both mammal diversity (S1 Table) and avian density (S3 Table). No significant differences were found in models that included vector variables (S2 and S4 Tables).

## Biodiversity components favouring or limiting the dilution effect

The number of different mosquito species captured at each sampling locality during the sampling period was positively related to the total number of mosquitoes captured at each locality (estimate ± S.E. = 1.649 ± 0.281, $t_{37.72}$ = 5.882, $p < 0.001$; Fig 4A). However, this relationship disappears when we consider the richness estimated from a rarefaction curve. Additionally, mosquito diversity was negatively associated with the total number of mosquitoes collected (-0.111 ± 0.028, $t_{43}$ = -3.977, $p < 0.001$), which was itself unrelated to both the density of avian hosts, and to the combined density of avian and mammal individuals. Rather, a negative relationship–and not a positive one as predicted–was found to exist between the total number of mosquitoes collected and the density of mammal hosts (-0.039 ± 0.012, $t_{32.82}$ = -3.243, $p < 0.003$; Fig 4B). Finally, while avian evenness was only marginally related to avian richness, calculated as the number of different bird species recorded at a locality (0.003 ± 0.002, $t_{43}$ = 1.814, $p = 0.077$), this relationship reached significance when avian richness was estimated from a rarefaction curve (0.015 ± 0.003, $t_{42.82}$ = 4.686, $p < 0.001$; Fig 4C).

## Discussion

Contrary to the predictions of DEH, none of the relationships between either host species diversity or richness were negatively associated with the prevalence of the four studied avian pathogens. Indeed, positive relationships were found between avian species richness and both WNV seroprevalence and *Leucocytozoon* prevalence, a pattern that is opposite to one expected. Moreover, when vector richness and evenness were incorporated into the system, the outcomes did not change. We would thus expect that any negative relationship between vertebrate

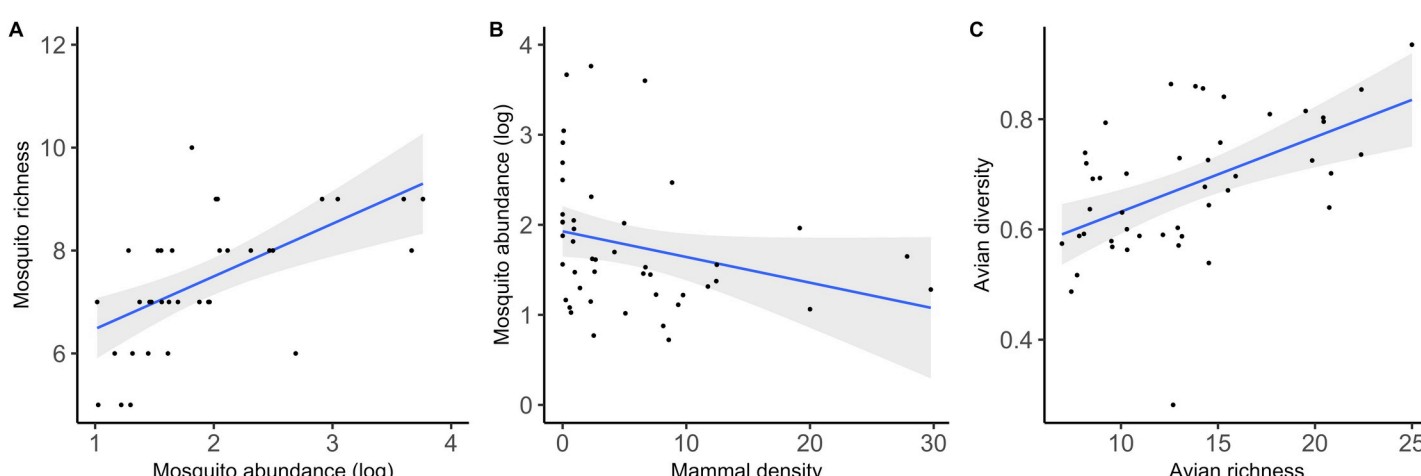

**Fig 4.** Relationships between A) mosquito richness (measured from a rarefaction curve) and the total number of mosquitoes collected; B) the number of mosquitoes collected and mammal density; and C) avian diversity (measured as the evenness index) and avian richness (estimated from a rarefaction curve) at the 45 localities included in this study. The 95% confidence level interval is shown in grey.

and mosquito community, and pathogen infection will depend on the local ecological factors that favour pathogen dilution. These results were not affected by age, sex, or seasonal changes in pathogen prevalence since these variables were included as individuals' covariates in the models.

The role played by the DEH is part of the intense debate that is currently raging regarding the ecosystemic services that biodiversity provides for public health [7,8,12,26,42,43]. The best support for the dilution effect comes from studies of Lyme disease [5,14,15,44–46], a number of which suggest that species diversity is negatively related to disease risk. However, as many authors have noted, separating the effects of biodiversity from those due to the presence of particular host species is problematic [19,42]. For example, the white-footed mouse (*Peromyscus leucopus*) may play a key role in disease epidemiology as it is highly efficient at infecting ticks and is thought to be the main natural reservoir of Lyme disease in eastern North America.

We tested DEH on a multi-pathogen system by taking into account both the potential role of vertebrate hosts and mosquito vectors. Although host diversity has been reported to generally inhibit pathogen transmission success [42], our results pose the critical question of how and how often the dilution effect operates in animal pathogens. We analysed the predictions of DEH in four different vector-borne pathogens and the expected negative relationship between parasite prevalence and avian host species richness or diversity was not found for any of the parasites. In fact, contrary to the predictions of DEH, WNV seroprevalence and *Leucocytozoon* infection were positively associated with the avian richness estimated from a rarefaction curve and we have no reason to think that the relationship would be any different for other avian species at the studied localities. Similarly, models including the number of different avian species at each locality reinforced these results with the addition of a significant and positive relationship with the prevalence of infection by *Haemoproteus* parasites. Overall, these results support the incidence of increased pathogen transmission in areas with richer avian communities, contrary to the predictions of DEH.

In North America, WNV infection rates in both mosquitoes and humans have been found to be negatively associated with non-passerine species richness [47]. Indeed, after controlling for socioeconomic factors potentially affecting the prevalence of WNV disease, a lower incidence of WNV in humans was found in US counties with greater avian species richness and diversity (i.e., evenness [10]). In addition, Allan et al. [48] found that WNV prevalence in vector mosquitoes was negatively related to avian diversity. By contrast, avian species richness was positively correlated with WNV seroprevalence in the areas we studied. Although the approaches used in these two studies differ, the contrasting patterns of WNV infection in relation to biodiversity may be linked to the different epidemiology of WNV on these two continents [49].

In southern Spain, even though the density of competent vectors such as the ornithophilic *Cx. perexiguus* may facilitate WNV transmission in birds [50], this species mostly occurs in natural rural areas and is less abundant near inhabited areas [51]. Thus, in our study area in Spain, flavivirus transmission relies mainly on the mosquito species that are most abundant in areas of high avian diversity [11,50], while in USA transmission is likely to be more linked to the mosquito species that frequent built-up areas [52], which are probably characterised by lower avian diversity. For instance, the mosquito species *Cx. perexiguus* seems to be responsible for much of the WNV transmission that occurs in southern Spain [50,53,54], where this species is commoner in less urban and more rural areas [51] possessing greater avian biodiversity than in urban areas. Conversely, in Spain, WNV circulates naturally between wild birds and mosquitoes [50] and only sporadic cases of humans with clinical symptoms are reported. Despite this, a large outbreak occurred in 2020 involving 77 human cases and seven deaths [55]. Usutu virus, another mosquito-transmitted flavivirus, is present in our study area and,

like WNV, has only been detected in SW Spain in *Cx. perexiguus* [11]. Similarly, this virus is mainly present in areas with high avian biodiversity, including the Doñana National Park [11].

WNV transmission could also be affected by the species composition of the vertebrate host community where one or a few avian species may be responsible for most transmission events [18]. Thus, differences between host species in terms of their exposure to mosquito bites and their competence for WNV transmission success may result in large local differences in the amplification of WNV linked to the composition of the avian community [18]. For these reasons we also checked for the potential effect of the phylogenetic diversity of the avian community. However, no associations were found with any of the four pathogens we investigated.

Differences in the feeding preferences of mosquitoes have been reported, with mosquito bites occurring more often in certain animal species [56] or in individuals with particular phenotypic characteristics (e.g., body size) [57]. For example, WNV mosquito vectors feed on American robins (*Turdus migratorius*) in North America and European blackbirds (*Turdus merula*) at a much higher rate than expected given their abundance in relation to other avian species. However, the blood-feeding patterns of mosquitoes may depend on the composition of the host community in the area [18,58]. This was reported by Kilpatrick et al. [18], who noted that an increase in WNV incidence in humans was linked to a shift in the feeding preferences of mosquitoes from birds to humans when the preferred host species, the migratory American robin, left for its winter quarters. Consequently, the potential relationship between biodiversity and pathogen prevalence may also be strongly influenced by the composition of vertebrate communities [11] and, in particular, by whether species-poor communities are dominated by competent or non-competent vertebrates. Mammal density in theory may also reduce pathogen prevalence in avian species as infected vectors biting mammals will not transmit the pathogen. However, mammal density was negatively related to *Haemoproteus* prevalence and positively associated to *Leucocytozoon* prevalence. At least in the case of biting midges, the main vector of *Haemoproteus*, mammals may provide alternative blood-feeding opportunities for insect vectors, thereby reducing parasite transmission success [59]. This is supported by the relatively opportunistic behaviour of important vectors of *Haemoproteus* such as *Culicoides circumscriptus*, which feeds on both mammals and birds [60]. However, according to our results, this may not be the case in blackflies, the main vectors of *Leucocytozoon*. Depending on their feeding patterns, this group is either ornithophilic or mammophilic due to differences in the structure of females' claws [61].

According to previous studies, an increase in vector diversity and richness may also favour pathogen transmission success [9,29]. In species-rich communities where there are more mosquito species, the probability that pathogens will interact with competent vectors may increase and compensate for the 'wasted' bites on reservoir species that are only weakly susceptible. However, this possibility was not supported by our results due to the lack of any association between the prevalence of *Plasmodium* and WNV seroprevalence and any measure of mosquito richness. *Plasmodium* and WNV are multi-vector pathogens, and a diversity of mosquito species are involved in their transmission [24,53,62], which underlines the complexity of the study system. In addition, Roche's model [29] assumed that the most abundant vector was also the most susceptible. Of the species sampled in our study, *Cx. perexiguus*, *Cx. pipiens* and *Cx. modestus*, the main vectors of WNV in Europe [62], were less abundant and widespread than other commoner species (i.e., *Ae. caspius* and *Cx. theileri*) that probably do not play such an important role in WNV transmission in the area [50,54].

Randolph and Dobson [12] have criticized the dilution effect since species-rich host communities may hold more individuals capable of sustaining a higher abundance of vectors, and so vertebrate-rich communities may in fact have higher pathogen transmission rates. Although this may be the case for ticks, for mosquitoes our results suggest that vector and host densities

are unrelated. In other words, in our study area, mosquito density is not limited by vertebrate availability but, rather, is probably more affected by other environmental factors such as climate and landscape characteristics [51,63]. The presence of suitable habitats for egg-laying and larval development may be a more serious limitation on mosquito populations than vertebrate densities since lower densities may be compensated for by increases in the biting rate per host [12]. Only in the case of mammals did we find a negative relationship between mammal and vector density, thereby suggesting that mammals may in fact avoid areas with more mosquitoes. Previous studies [12,29] have been based on assumptions that were not supported by our study, namely, i) host community assembly is additive and ii) the density of vectors and reservoirs are related.

Quantifying biodiversity is a difficult task since the methods used may bias estimates of species richness, diversity, and density. However, our results show that the lack of any negative relationship between biodiversity and pathogen prevalence do not depend on the diversity estimators employed. In addition, some methodological limitations may affect the conclusions obtained. For example, the method used here to record avian species was obviously biased against nocturnal species such as nightjars and owls, while our mammal estimates were biased against rodents. However, our studies of mosquito diets in the area suggest that these other groups account for only a very small fraction of mosquito bloodmeals [54] and, consequently, will not have any critical effect on the studied pathogen transmission. Furthermore, despite the large sample size, we only studied pathogen prevalence in a single avian species. However, this was the only species present at all the studied localities and by focussing on just one species we were able to compare pathogen prevalence between different localities. Interestingly, house sparrows are bitten by *Cx. pipiens* at a frequency similar to that expected given their relative abundance in the avian community [56] and consequently should be a reliable avian species for surveying pathogens such as *Plasmodium* and WNV transmitted by this mosquito species [53]. In addition, even though avian malaria parasites vary in their ability to infect different avian species [64], some of the parasites infecting house sparrows are generalists. This is the case of the *Plasmodium relictum* lineage SGS1 infecting house sparrows [65] that has been found to infect more than 125 species belonging to a number of different orders [66]. This may also be the case in WNV, which is a multi-host multi-vector pathogen able to infect more than 300 species of birds (https://www.cdc.gov/westnile/dead-birds/index.html) according to a wide range of experimental studies [37] as WNV antibodies have been identified in many different bird species in the study area [67]. Thus, although some specialist parasite-host assemblages do occur in this community, most of the pathogens studied here are not restricted to house sparrows, which probably reflects the general pattern shown in other species present in the area. Therefore, the patterns found in house sparrows may reflect those occurring in other species in the area, and we have no reason to think that this bird species is an exception in this community. It is important to note that the dilution effect refers to a reduction in disease transmission in the system and consequently should be detectable in any host species present.

## Conclusions

By characterizing mosquito and vertebrate communities present at different localities with diverse biotic and abiotic conditions, this work simultaneously analyzed the influence of biodiversity on four pathogens that use multiple vector species for their transmission. We found no support for DEH as a general process operating on vector-borne pathogens, which suggests that any relationship between host and/or vector biodiversity and pathogen prevalence will depend heavily on host community composition and the characteristics of the pathogen. Although our study system possessed many qualities that make it a good candidate for the

occurrence of the dilution effect (i.e., a vector-borne pathogen, an area where vector and host density are unrelated, and substitutive community assembly; see [21,25]), the relationship between avian species richness and pathogen prevalence was non-significant for one of the pathogens and positive for the other three. Our results suggest that many of the assumptions made by models analysing the viability of the dilution effect are unrealistic or, at least, not applicable to our study system. The dilution effect may operate locally under certain circumstances (specific areas and/or diseases) but, as our results suggest, it cannot be regarded as an emerging property of biodiversity (see for example [68]). Consequently, the range of pathogens studied needs to be broadened and a 'One Health' approach applied to fully understand the ecological factors that favour pathogen dilution and the frequency with which these conditions occur in nature.

## Materials and methods

### Ethics statement

The CSIC Ethics Committee approved the experimental procedures on 9 March 2012. This study did not affect any endangered species.

Mosquito and bird trapping were carried out with all the necessary permits from the Consejería de Medio Ambiente, and Consejería de Agricultura, Pesca y Desarrollo Rural (Junta de Andalucía). Entomological surveys and bird sampling on private land and in private residential areas were conducted with all the necessary permits and consent, and in the presence of owners.

Fieldwork was conducted in 2013 in southern Spain, an area of Mediterranean climate with long dry summers and most rainfall in winter. The study was carried out in 45 localities in Cadiz, Huelva, and Seville provinces, which were grouped into triplets (Fig 5) of habitat category (urban, rural, and natural) to maximize differences in biodiversity whilst controlling for geographically structured factors (see *Statistical analyses*). The three localities in each triplet were visited to capture insect vectors or count vertebrates on the same day, while house sparrows were sampled in different field sessions on consecutive days at sites within the same triplet. The median delay between the vertebrate census and mosquito sampling was 0 days, with a 25% quantile of six days before and a 75% quantile of nine days afterwards.

### Bird and mosquito sampling

House sparrows were captured using mist nets at the 45 localities in July–October after the breeding season to facilitate the capture of juvenile birds that would be the best test of pathogen circulation in that year. Birds were marked, sexed, and aged [69], and a blood sample was taken from the jugular vein of each bird before immediate release at the place of capture.

Data from the mosquito captures has previously been analysed by Ferraguti et al. [51] to identify the impact of landscape anthropization on mosquito communities (see S1 Text for further information on vector sampling and community abundance and composition). In brief, mosquitoes were captured in April–December 2013, corresponding to the period of maximum mosquito activity in southern Spain [63]. Mosquitoes were preserved on dry ice and then transported to the laboratory for identification to species level [24]. Mosquitoes belonging to the *univittatus* complex were identified as *Culex perexiguus* based on male genitalia, following Harbach [70].

For each locality, we calculated i) the mosquito species richness estimated from a rarefaction curve with the function *rarefy* (package *vegan*, [38]); ii) the diversity of mosquito communities calculated as the evenness, i.e., the similarities between the frequencies of individuals belonging to the different species that constitute a community using Shannon's equitability

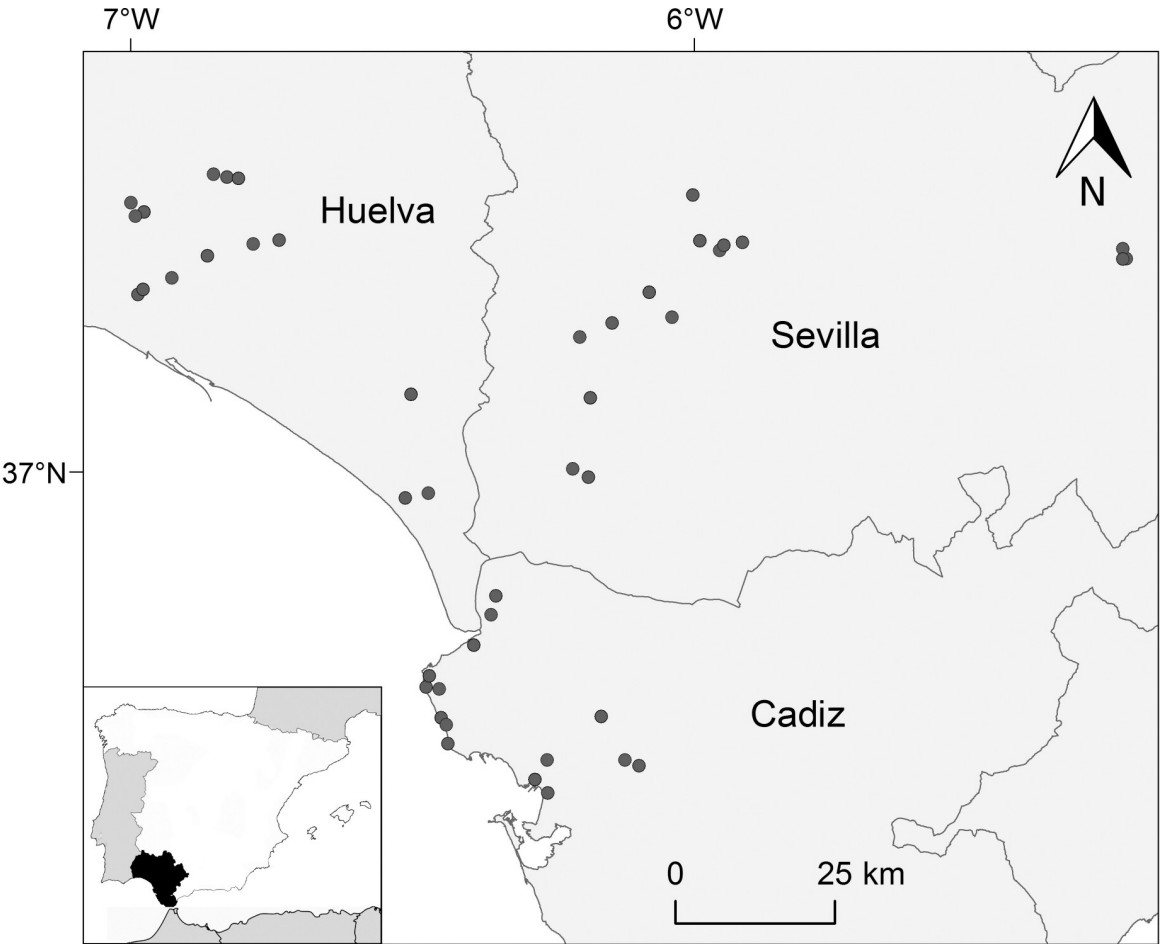

**Fig 5. Distribution of the 45 localities at which house sparrows were captured in southwest Spain.** Map was built with ArcGIS v10.2.1 (ESRI, Redland) and developed by using shape files of Datos Espaciales de Referencia de Andalucía (DERA, https://www. juntadeandalucia.es/institutodeestadisticaycartografia/DERA/g13.htm).

index [39]; and iii) the number of total captures of each mosquito species. The mean value of the daily number of mosquitoes captured was calculated for each of the 45 localities to test for the potential relationships in the community additive assembly model for vectors assumed by Roche et al. [29] and the association between host and vector density proposed by Randolph & Dobson [12]. Additionally, further models were conducted using the number of different mosquito species registered at each locality during the whole sampling period. The number of mosquitoes sampled was used as an estimate of mosquito abundance/density assuming that the sampled area was equivalent throughout the season and at all localities.

## Vertebrate censuses

Avian and mammal counts were conducted in June–November 2013 at the same localities as the mosquito sampling. Although the vertebrate community could vary between seasons (i.e., due to the arrival of migrant individuals), we included the mean value of the vertebrate censuses conducted in June–November that coincided with the house sparrow captures (see S1 Text for further information on vertebrate censuses and community abundance and composition).

For each locality, we calculated the average values for the summer and autumn counts of avian and mammal densities, species richness estimated from a rarefaction curve [20,38], and species diversity calculated as the evenness using Shannon's equitability index [39]. Alternative models were also conducted using the number of different vertebrate species recorded at each locality, while the avian phylogenetic diversity was implemented in avian community models [40,41]. The average value for summer and autumn counts of the sum of the vertebrate density (avian plus mammal) was estimated for each of the 45 localities to test for the association between host and vector density proposed by Randolph & Dobson [12]. For further details on vertebrate community abundance and composition, see Ferraguti et al. [65].

## Molecular and serological analyses

Genomic DNA was extracted from blood samples and the cell fractions of all house sparrows sampled using the Maxwell16 LEV system Research (Promega, Madison, WI). Infections by *Plasmodium*, *Haemoproteus* and *Leucocytozoon* parasites were detected following Hellgren et al. [71]. Molecular analyses of negative samples were repeated to avoid false negatives [72]. Both negative controls for PCR reactions (at least one per plate) and DNA extraction (one per 15 samples) were included in the analysis. Positive amplifications were sequenced using the Macrogen sequencing service (Macrogen Inc., The Netherlands) to identify the parasite genus. Sequences were identified by comparing with the GenBank DNA sequence (National Center for Biotechnology Information, Blast). The *Plasmodium*, *Haemoproteus* and *Leucocytozoon* parasites infecting the birds studied here are described in Ferraguti et al. [65] and Jiménez-Peñuela et al. [73].

Bird sera were screened to detect antibodies against WNV with the ELISA kit INGEZIM West Nile COMPAC (Ingenasa Spain) [74]. Not enough serum was obtained for 44 out of the 2,588 birds sampled, so these individuals were excluded from WNV analyses. Positive or doubtful ELISA samples were subsequently analysed with a virus neutralization test (VNT) using the micro-assay format (96-well plates), as described by Llorente et al. [75]. Neutralizing antibody titres were determined in parallel for each serum sample against WNV (strain Eg-101) and Usutu virus (USUV, strain SAAR1776) using serial (twofold) dilutions (1:10–1:1280) of each serum sample in a VNT. Observed neutralizing immune responses were considered specific for WNV when VNT titres were at least four-times higher than for USUV. USUV, belonging to the Japanese encephalitis group, is another flavivirus currently circulating between birds and mosquitoes in the area [67,75]. Usutu prevalence in house sparrows (0.04%) was too low to allow for any analysis of the relationship between Usutu prevalence and biodiversity. Data on the prevalence of WNV antibodies in these birds has previously been analysed by Martínez-de la Puente et al. [50] to identify the main vectors involved in WNV transmission and the risk of spillover to humans.

## Statistical analyses

Firstly, Generalized Linear Mixed–Effects Models (GLMM) with a 'logit' link function and binomial distribution were used to investigate which factors are associated with infection by *Plasmodium*, *Haemoproteus* and *Leucocytozoon*, and WNV seroprevalence in wild house sparrows. Separate models were used for each pathogen. The infection status of each individual (infected or uninfected) for each pathogen was included as the dependent variable, while bird age and sex (categorical), month of capture, and avian and mammal density, richness and diversity were included as continuous independent variables. Independent models were performed for the different metrics of species richness and diversity. Individual variables such as age, sex and month were included to control for potential differences in prevalence of

pathogens between less than one-year-old birds, males and females, and the increasing prevalence of pathogens as the summer progressed [65,76,77].

In addition to the host variables, for the mosquito-borne pathogens *Plasmodium* and WNV, additional GLMMs including mosquito species richness (continuous) and evenness (continuous) were performed. *Haemoproteus* and *Leucocytozoon* prevalence analyses were restricted to the vertebrate community as we lacked information on the density of their main vectors (*Culicoides* and blackflies, respectively). Province, triplet nested in province, and locality nested in triplet and province, were included as random factors to account for the geographical stratification of the sampling design. For each GLMM, the marginal (considering only fixed factors) and conditional (considering fixed and random factors) variance explained ($R^2$) were calculated following Nakagawa and Schielzeth [78]. The collinearity between all independent variables was tested using the Variance Inflation Factor (VIF) [79]; GLMM overdispersion was checked for using the Pearson statistic (ratio of the Pearson $\chi^2$ to its degrees of freedom), a common method for assessing the deviance of goodness-of-fit statistics [80]. We found no evidence of collinearity between the variables included in the models or of overdispersion, as the Pearson dispersion statistics were always close to 1.

Secondly, LMMs were used to test biodiversity components favouring or limiting the dilution effect by analysing the relationships between mosquito and vertebrate (avian and mammal) variables estimated at each of the 45 localities. Normality of continuous variables and the residuals of all the LMMs were tested by checking normality *qq-plots* and Shapiro-Wilk's normality tests. The number of total mosquito captures (continuous, mean number of mosquitoes trapped per day in the locality) was log-transformed to normalize their distribution. Specifically, independent LMMs were performed to test the relationship between mosquito richness (dependent) or diversity (dependent) and the number of total captures (independent), the number of mosquito total captures (dependent) and the densities of i) avian hosts, ii) mammal hosts and iii) total vertebrates, as independent variables. Finally, the association between avian diversity (dependent) and avian richness (independent) was also tested. The 95% confidence intervals (C.I.) of *Plasmodium*, *Haemoproteus* and *Leucocytozoon* and the seroprevalence of WNV were calculated with the function *binconf* from the package Hmisc.

All statistical analyses were conducted in R [81] using the packages *arm*, *car*, *ggplot2*, *lme4*, *MASS*, *Matrix*, *MuMIn*, *Rcpp*, *stats* and *vegan*. The database used for the statistical analyses and the numerical data used in all figures are included in S1 Data.

## Supporting information

**S1 Table. Results of the GLMMs testing the relationships between the prevalence of avian malaria *Plasmodium*, the related *Haemoproteus* and *Leucocytozoon* parasites (N = 2,588), and the and the seroprevalence of WNV (N = 2,544) and individual characteristics of house sparrows (age, sex, and month of capture), avian and mammal species density, richness (estimated as the raw number of different avian or mammal species registered at each sampling site), and diversity (calculated as evenness index).** Significant relationships ($p \leq 0.05$) are highlighted in bold; conditional and marginal (in brackets) $R^2$ variance are shown.
(PDF)

**S2 Table. Results of the GLMMs analysing the relationship between the prevalence of the two studied mosquito-borne pathogens: avian malaria *Plasmodium* (N = 2,588) and seroprevalence of WNV (N = 2,544) and the individual characteristics of the house sparrows (age, sex, and month of capture), avian and mammal species density, richness (measured from the raw number of different avian or mammal species registered at each sampling**

site) and diversity (calculated as evenness index), and vector species richness (measured as the raw number of different mosquito species captured at each sampling) and diversity (calculated as evenness index). Significant relationships ($p \leq 0.05$) are highlighted in bold. Conditional and marginal (in brackets) $R^2$ variance are shown.
(PDF)

**S3 Table. Results of the GLMMs testing the relationships between the prevalence of avian malaria *Plasmodium*, the related *Haemoproteus* and *Leucocytozoon* parasites (N = 2,588), and the seroprevalence of WNV (N = 2,544), and individual characteristics of house sparrows (age, sex, and month of capture), avian species density, richness (estimated as the raw number of different avian species registered at each sampling site) and diversity (estimated as avian phylogenetic diversity), mammal species density, richness (measured from the raw number of different mammal species registered at each sampling site) and diversity (calculated as evenness index).** Significant relationships ($p \leq 0.05$) are highlighted in bold; conditional and marginal relationships are in brackets; $R^2$ variance are shown.
(PDF)

**S4 Table. Results of the GLMMs analysing the relationship between the prevalence of the two mosquito–borne pathogens studied: avian malaria *Plasmodium* (N = 2,588) and seroprevalence of WNV (N = 2,544) and the individual characteristics of the house sparrows (age, sex and month of capture), avian species density, richness (measured from the raw number of different avian species registered at each sampling site) and diversity (calculated as the avian phylogenetic diversity), mammal species density, richness (measured from the raw number of different mammal species registered at each sampling site) and diversity (calculated as evenness index), and vector species richness (measured from the raw number of different mosquito species captured at each sampling) and diversity (calculated as evenness index).** Significant relationships ($p \leq 0.05$) are highlighted in bold. Conditional and marginal (in brackets) $R^2$ variance are shown.
(PDF)

**S1 Data. Excel spreadsheet containing in separate sheets the numerical data used for the statistical analysis and for the figure preparation.**
(XLSX)

**S1 Text. Supporting information: methods and results.**
(PDF)

## Acknowledgments

Alberto Pastoriza, Manuel Vázquez, Manolo Lobón, Óscar González, Carlos Moreno, Cristina Pérez, Esmeralda Pérez, Juana Moreno Fernández, Antonio Magallanes and Martín de Oliva helped with the fieldwork and mosquito identification. Isabel Martín, Laura Gómez, Francisco M. Miranda Castro, Olaya García Ruiz, Carmen Barbero Ameller and Antonio Sanz (INgenasa) collaborated with the laboratory analyses. We are grateful to all the landowners and to the Consejería de Medio Ambiente for allowing us to work on their properties. Mike Lockwood revised the English text.

## Author Contributions

**Conceptualization:** Martina Ferraguti, Josué Martínez-de la Puente, Jordi Figuerola.

**Data curation:** Martina Ferraguti, Josué Martínez-de la Puente, Jordi Figuerola.

**Formal analysis:** Martina Ferraguti.

**Funding acquisition:** Ramón Soriguer, Jordi Figuerola.

**Investigation:** Martina Ferraguti, Josué Martínez-de la Puente, Miguel Ángel Jiménez–Clavero, Francisco Llorente, Jordi Figuerola.

**Methodology:** Martina Ferraguti, Josué Martínez-de la Puente, Miguel Ángel Jiménez–Clavero, Francisco Llorente, David Roiz, Jordi Figuerola.

**Project administration:** Ramón Soriguer, Jordi Figuerola.

**Resources:** Ramón Soriguer, Jordi Figuerola.

**Supervision:** Josué Martínez-de la Puente, Ramón Soriguer, Jordi Figuerola.

**Validation:** Martina Ferraguti, Francisco Llorente.

**Visualization:** Martina Ferraguti.

**Writing – original draft:** Martina Ferraguti.

**Writing – review & editing:** Martina Ferraguti, Josué Martínez-de la Puente, Miguel Ángel Jiménez–Clavero, Francisco Llorente, David Roiz, Santiago Ruiz, Ramón Soriguer, Jordi Figuerola.

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
