## [Decision Letter · Decision Letter 0]

1 Mar 2021

Dear Dr. Ferraguti,

Thank you very much for submitting your manuscript "A field test of the dilution effect hypothesis in four avian multi–host pathogens" for consideration at PLOS Pathogens. As with all papers reviewed by the journal, your manuscript was reviewed by members of the editorial board and by several independent reviewers. The reviewers appreciated the attention to an important topic. Based on the reviews, we are likely to accept this manuscript for publication, providing that you modify the manuscript according to the review recommendations.

Dear Dr. Ferraguti,

Two experts have now reviewed your resubmitted work “A field test of the dilution effect hypothesis in four avian multi–host pathogens” (PPATHOGENS-D-21-00275). These are the same scientists that reviewed previous version of this work.

While Reviewer 2 is happy with the new version, Reviewer 1 still has relevant concerns regarding the presentation of ecological concepts and how they relate to the objectives and results of the study. He/she also underlines that the text is often difficult. Reviewer 1 indicates that all his/her concerns ca be solved in a rewrite of the text, and in many of the very detailed minor comments provided he/she proposes solutions. I want to underscore that the English needs to be thoroughly revised as in my opinion, and despite M. Lockwood, it is often imprecise and contains grammatical and idiomatic errors.

In addition to the reviewer comments, the authors sould also consider:

Lines 110-11: the relationship diversity-disease risk may also depend on the scale of the study.

Discussion: Because some of the pathogens studied infect other species, in which prevalence was not analysed, I would like to see a comment on the discussion about how prevalence in other hosts will affect the interpretation of the results.

I agree with Reviewer 1 and with previous comments fo Reveiwer 2, in that this is a study based on a rather unique data base, presenting relevant results and conclusions, and I think that if all the comments of Rev. 1 and mine are addressed it will make a very good paper.

Sincerely

Sincerely,

Fernando Garcia-Arenal

Guest Editor

PLOS Pathogens

Kirk Deitsch

Section Editor

PLOS Pathogens

Kasturi Haldar

Editor-in-Chief

PLOS Pathogens

orcid.org/0000-0001-5065-158X

Michael Malim

Editor-in-Chief

PLOS Pathogens

orcid.org/0000-0002-7699-2064

Dear Dr. Ferraguti,

Two experts have now reviewed your resubmitted work “A field test of the dilution effect hypothesis in four avian multi–host pathogens” (PPATHOGENS-D-21-00275). These are the same scientists that reviewed previous version of this work.

While Reviewer 2 is happy with the new version, Reviewer 1 still has relevant concerns regarding the presentation of ecological concepts and how they relate to the objectives and results of the study. He/she also underlines that the text is often difficult. Reviewer 1 indicates that all his/her concerns ca be solved in a rewrite of the text, and in many of the very detailed minor comments provided he/she proposes solutions. I want to underscore that the English needs to be thoroughly revised as in my opinion, and despite M. Lockwood, it is often imprecise and contains grammatical and idiomatic errors.

In addition to the reviewer comments, the authors sould also consider:

Lines 110-11: the relationship diversity-disease risk may also depend on the scale of the study.

Discussion: Because some of the pathogens studied infect other species, in which prevalence was not analysed, I would like to see a comment on the discussion about how prevalence in other hosts will affect the interpretation of the results.

I agree with Reviewer 1 and with previous comments fo Reveiwer 2, in that this is a study based on a rather unique data base, presenting relevant results and conclusions, and I think that if all the comments of Rev. 1 and mine are addressed it will make a very good paper.

Sincerely

Reviewer Comments (if any, and for reference):

Reviewer's Responses to Questions

**Part I - Summary**

Reviewer #1: The manuscript brings together a number of interesting components of the dilution effect hypothesis (DEH), and is a timely and methodologically sound contribution. The prevalence of four pathogens estimated from avian species, is regressed against a number of explanatory variables that are expected to reveal a mechanistic understanding of the DEH. Prevalence in avian species is related to the richness, diversity, density, phylogenetic diversity, and evenness of host and vector species assemblage. The results showed a number of inconsistencies with key assumptions of the dilution effect hypothesis. The key finding was that the prevalence of most of the avian pathogens studied, exhibited a positive relationship with community parameters used commonly in DEH research. The study is novel and of broad interest because empirical data derived from a tractable study system is used to get at the mechanics of the relationship between biodiversity and disease risk.

Reviewer #2: I have reviewed a previous version of this manuscript. The authors have adequately addressed my comments and the manuscript has been substantially improved. I have no further comments.

**Part II – Major Issues: Key Experiments Required for Acceptance**

Reviewer #1: The manuscript covers a lot of ground by introducing many concepts developed by others, to explain mechanics of the DEH. At times, the reading was difficult work for me, and also likely for the uninitiated virologist not savvy in the terminology of community ecology. This comes about primarily due to the vagueness or incomplete explanation for key concepts introduced at the beginning of the manuscript. These issues can be rectified in a rewrite that provides specific references (e.g. point to, and give details of, the particular key finding of interest presented in a given reference), and explains more thoroughly the cited work already given. In particular, I found the premise of, and explanation for, additive and subtractive assembly incomplete, and difficult to integrate with the other concepts presented in the introduction, results, and discussion. These are major concerns. Specifics of minor concerns are given below.

Reviewer #2: (No Response)

**Part III – Minor Issues: Editorial and Data Presentation Modifications**

Reviewer #1: Line 31: Suggestion. A succinct statement delineating the relevance of density- and frequency-dependent transmission in relation to pathogen-host encounters and the dilution effect hypotheses, would provide a general basis for developing the subsequent arguments that rely on variation in species density.

Line 49: ‘possessing’.

Line 52: Be specific when describing variables of relationships that were found to important. What parts of ‘biodiversity’ were regressed here. As the strength of the manuscript is its mechanistic perspective on the DEH, a conclusion commenting on this theme would be good; i.e., how did community composition, species identity, and abundance influence prevalence in avian hosts.

Line 89: Expression; “…reported in the wild”, or, have been observed in wild populations?

Line 96. “proposed by the dilution effect”, or, in the DEH…

Line 110: Non-experts may not know what community composition refers to, and how it differs from diversity (and structure). For example, composition typically separates the species identity and species abundance components of diversity.

Lines 117 to 128: I read this part several times, and the connection between the hypotheses of Rohr et al. (2020), and the analyses conducted in the manuscript are unclear. Rohr et al. (2020) states "When community (dis) assembly is substitutive, amplification can occur when the addition of individuals of new, competent host species reduce the density of less competent host species. Amplification or dilution can occur when competent hosts or non-competent hosts, respectively, are added to or subtracted from communities via the sampling effect (that is, more diverse communities are more likely to contain a host species that either strongly increases or decreases disease)". The authors provide little information (and no empirical data that I can see) on whether the addition of non-competent or competent hosts occurs in the study system.

Lines 145 to 142: This statement is not entirely correct. The Roche et al. (2013) model assumed that the most abundant vector and reservoir species had the highest susceptibility (i.e., not competence). In their model, competence was "modulated" (i.e., correlated) by susceptibility. The difference between susceptibility and competence is not trivial and the differences deserve attention because it may alter how the conclusions of the study are interpreted.

Line 149: The authors introduce the term ‘assembly’ that relates to successional pathways and the temporal changes in biological communities. This is clear enough, and the references to Rohr et al. (2020) set the conceptual background for expectations given either additive or subtractive assembly. However, I am not sure how these hypotheses fit with the analyses presented in the manuscript. I struggled to see a connection between the objectives of the study, broadly to look at avian prevalence as explained by the composition of the vector and host assemblages, and the hypotheses for subtractive and additive assembly. A rewrite should include a better explanation of this connection, if indeed one is intended.

Line 176: What are “all the factors”? Understanding “all the factors” is not likely.

Lines 183 to 185: I do not understand the meaning of this sentence.

Line 186: What do the authors mean by “evenness analysis for richness and diversity parameters”? Evenness is a parameter, not a type of analysis.

Lines 194 to 199: As I have understood the "additive" and "subtractive" hypotheses of assembly in respect to the relative proportions of competent to non-competent hosts (as presented in the background above: Rohr et al. 2020), I do not know how measuring correlations (e.g. vector richness vs. vector density) informs on the numbers of competent and non-competent hosts (or high/low susceptibility hosts), or whether this proportion, or the phenotype introduced during community assembly, is important to the direction of the manuscript. How does the correlation inform on the subtractive and additive hypotheses for assembly, and then, how do the mode(s) of assembly relate to the study system?

In the Methods, Lines 464 to 467; i.e., the mean (species?) relative (or total?) abundance of mosquitoes captured each day was calculated. How exactly does daily variation in the mean abundance inform on successional changes in the mosquito assemblage? And, how does it inform on the proportions of competent (i.e., to transmit) to non-competent host phenotypes? The references given for the work of Rohr et al. (2020) and Randolph and Dobson (2012) should direct the reader to specific aspects of each of these studies. I read them, and could not find an obvious connection. The work by Randolph and Dobson (2012) refers to theoretical (i.e., "in principle") outcomes of adding non-competent hosts to an assemblage, so the phenotype is known or assumed in this case. Given that the phenotype (i.e. non-competent) is assumed/known, decreases in encounters with other hosts [“humans”] can be expected. The absolute number of vectors is important, [“not the proportion of vectors infected”], but under the assumption that non-competent hosts have been added. Here, I am lost.

Line 200 to 204: Why are there no references to prevalence as the dependent variable? The Results section begins with an analysis of prevalence. It is not clear at this stage of the manuscript what the objectives of the study are, nor the hypotheses being tested; because there are no explicit statements to this effect.

Lines 210 to 232: The data are impressive, and the motivation to analyse the composition of assemblages as well as successional pathways is a very novel aspect of the manuscript. The introduction argued that community composition (taxonomic identity and their abundances) and diversity were important to separate, in understanding disease risk. This proposition is a strong component of the manuscript, but the regressions include largely independent variables for richness and diversity. Only one model speaks to composition (i.e., evenness). The results that point to the regression modelling are clear, but I am not sure what information I am supposed to gain from the other results reported here. Are they for the purposes of understanding the assembly hypotheses? Are the results of rarefaction of secondary importance? Rarefaction is typically used to address uncertainty in sample completeness and can be compared to the values of the raw observations in a Table somewhere, usually in the Supplementary Information. If the rarefaction results and mean values are of primary importance, the information would be easier to digest if it were presented in a compact, summary form that informs on their purpose. Do the mosquito abundance results and those referenced in the Ferraguti et al. (2016, 2017) papers, comprise the main comment on community composition made in the manuscript?

Lines 214 to 215: What does this mean: “and a rarefaction value of

4.54 (range 1.99 – 6.93), in each locality”? It reads like the same mean value was found in each locality.

Line 225: Another example of poor expression: “Vertebrate counts reported the total presence of…” Vertebrate counts do not report.

Line 236 and 247: Tables 1 and 2 include independent variables demographic variables (i.e., the population level), but I do not know its relevance to what has been introduced thus far. How do the demographic parameters fit into the general objectives and hypotheses?

Line 250: …models that included…

Line 260: Is this a regression of species richness and total abundance (i.e., total number of mosquitoes collected) at each site? This sentence appears to contradict the former.

Line 261 to 262: Is “site” the same as “locality”?

Line 281: It would be great to have a concise summary of the main finding in respect to how mechanics related to community composition influenced prevalence in the study system: i.e., “…separate the effects of biodiversity from those due to the presence of particular host species”. Initially in the discussion, there are many references to the work of others, but few in respect to the findings presented in the manuscript. The main findings appear to relate to the positive relationships between avian prevalence and parameters such as richness and diversity. This type of result does not live up to the direction of the study laid out in the introduction: “pathogen prevalence may be due to either the presence of key species or to the community composition rather than to any intrinsic property of biodiversity”. From that point in the introduction, the exciting topic of community assembly is introduced, thus raising the expectation of the reader beyond the typical disease-diversity paradigm. A change of emphasis in the discussion would help point out where the manuscript contributed most strongly to the field.

Line 298: In its current state, the main finding appears to be: “these results support the incidence of increased pathogen transmission in areas with a richer avian community contrary to the expectations of the DEH.” With all the analyses and results given about other hosts and vectors, is there not a main finding that can be related to the composition of the various assemblages?

Line 313 and 314: This relationship is interesting because it relates to composition (taxonomic identity and abundance). Are there any results that support this speculation?

Line 314: Is “community structure” supposed to be “community composition” here? The structure of a community is influenced by abiotic variables and so does not fit the context of the sentence.

Line 336: …than other species.

Lines 344 to 345: Which relationship exactly is being referred to? Is it prevalence in avian species as predicted by mammal density?

Lines 368 to 371: Randolf and Dobson (2012) frame infections rates in terms of vector abundance, not vector density, and host density. If vector density is not correlated with vector abundance, how is the comparison with the referenced work valid?

Line 381: It is not apparent to me how the study showed additive community assembly.

Lines 403 to 404: “pattern” or patterns; “those” or that?

Line 410: analysed

Line 418: I did not find information about substitutive assembly in these two references.

Line 536: I see no references in the introduction about why demographic variables are important predictors of risk.

Line 541: Richness is ordinal (unless rarefied).

Reviewer #2: (No Response)

PLOS authors have the option to publish the peer review history of their article (what does this mean?). If published, this will include your full peer review and any attached files.

Reviewer #1: No

Reviewer #2: No

Figure Files:

Data Requirements:

Reproducibility:

References:

---

## [Editor Report · Decision Letter 1]

12 May 2021

Dear Dr. Figuerola,

We are pleased to inform you that your manuscript 'A field test of the dilution effect hypothesis in four avian multi-host pathogens' has been provisionally accepted for publication in PLOS Pathogens.

Best regards,

Fernando Garcia-Arenal

Guest Editor

PLOS Pathogens

Kirk Deitsch

Section Editor

PLOS Pathogens

Kasturi Haldar

Editor-in-Chief

PLOS Pathogens

orcid.org/0000-0001-5065-158X

Michael Malim

Editor-in-Chief

PLOS Pathogens

orcid.org/0000-0002-7699-2064

All comments made to the original version of the manuscript by reviewer 1 and by myself have been properly addressed by the pertinent modifications of the text. In my opinion the paper is much improved and much more readable. I think this study makes an important contribution.
---

## [Editor Report · Acceptance letter]

8 Jun 2021

Dear Dr. Figuerola,

We are delighted to inform you that your manuscript, "A field test of the dilution effect hypothesis in four avian multi-host pathogens," has been formally accepted for publication in PLOS Pathogens.

Best regards,

Kasturi Haldar

Editor-in-Chief

PLOS Pathogens

orcid.org/0000-0001-5065-158X

Michael Malim

Editor-in-Chief

PLOS Pathogens

orcid.org/0000-0002-7699-2064